# Frontiers in Bioengineering and Biotechnology: Plant Nanoparticles for Anti-Cancer Therapy

**DOI:** 10.3390/vaccines9080830

**Published:** 2021-07-28

**Authors:** Erum Shoeb, Uzma Badar, Srividhya Venkataraman, Kathleen Hefferon

**Affiliations:** 1Department of Cell and Systems Biology, University of Toronto, Toronto, ON M5S 1A1, Canada; erumsh@uok.edu.pk (E.S.); uzmabadar.ca@gmail.com (U.B.); byokem@hotmail.com (S.V.); 2Department of Genetics, University of Karachi, Karachi 75270, Pakistan

**Keywords:** nanoparticles, plant virus-like particles, therapeutics, imaging, cancer

## Abstract

Naturally occurring viral nanomaterials have gained popularity owing to their biocompatible and biodegradable nature. Plant virus nanoparticles (VNPs) can be used as nanocarriers for a number of biomedical applications. Plant VNPs are inexpensive to produce, safe to administer and efficacious as treatments. The following review describes how plant virus architecture facilitates the use of VNPs for imaging and a variety of therapeutic applications, with particular emphasis on cancer. Examples of plant viruses which have been engineered to carry drugs and diagnostic agents for specific types of cancer are provided. The drug delivery system in response to the internal conditions is known as stimuli response, recently becoming more applicable using plant viruses based VNPs. The review concludes with a perspective of the future of plant VNPs and plant virus-like particles (VLPs) in cancer research and therapy.

## 1. Introduction

One of the great challenges in biomedicine and cancer treatment is how to specifically target drugs to tumor cells and increase therapeutic outcomes. Virus-based nanotechnology holds great promise for the design of targeted therapeutics, by taking advantage of the natural properties of viruses [1,2]. Recently, viral nanoparticles (VNPs) have attracted considerable attention as a unique class of nanocarriers for biomedical applications [3]. In addition to their ease of production and quality control maintenance, plant virus VNPs offer a logical alternative to synthetic nanoparticles as they are both biocompatible and biodegradable. 

For biotechnological applications, plant viruses have emerged as promising tools to be used as nanomaterials. Features which distinguish plant viruses from synthetic nanocarriers include stability, flexibility, diversity in shape and size for use in drug delivery and the nontoxic nature of plant viruses in humans. The revolutionary idea of producing virus-like particles (VLPs), which are empty capsids devoid of any genetic material and are replication-deficient, has been employed. VLPs acting as nanoparticles have the ability to encapsulate therapeutic molecules within capsids for imaging as well as for several drug delivery applications [4]. 

Nanoparticles have been further improved for their performance in terms of stimuli-responsivity, as viruses are naturally adapted to respond to stimuli. The tumor microenvironment is not like normal cells in terms of having acidic pH, or changed redox potential, justifying the application of a stimulus-sensitive tumor targeting nanoparticles or imaging delivery systems [5]. In order to provide controllable vectors, stimulus-responsive viruses capable of responding to variable environmental conditions are being developed for therapies of a number of diseases including cancer [6].

Cancer is one of the most common death causing disease worldwide and it is characterized by uncontrolled rapid cell division and differentiation [7]. The lack of a suitable drug delivery system is the main cause of failure as far as treatment is concerned despite the availability of several chemotherapeutic agents. Therefore, an efficient delivery systems is the real requirement, which would improve the delivery of antitumoral drugs at the target affected site to minimize the unwanted effects on healthy cells and maximize the efficiency of treatment [8]. VNPs are an ideal choice to apply for cancer treatment owing to the enhanced permeation and retention (EPR) potential of cancer cells for these nanoparticles, whereas VNPs cannot penetrate through healthy tissues due to tightly packed endothelial cells [9].

Plant viruses used for nanoparticles fall into the category of rod shaped, such as Tobacco mosaic virus (TMV) (Figure 1) and Potato virus X (PVX), or icosahedral shaped, such as Cowpea mosaic virus (CPMV) and Cowpea chlorotic mottle virus (CCMV). Different shaped viruses have different aspect rations and respond differently as nanoparticles in vivo, for example, in a tumor environment, as will be discussed later in this review. Both categories also possess unique features; for example, Tobacco mosaic virus can carry a drug payload on the surface or to a limited extent, within the inner channel of the nanoparticle in the absence of its RNA genome. TMV can also be coaxed to form spheroid nanoparticles. Potato virus X, on the other hand, cannot self-assemble in the absence of its RNA genome, and thus can only carry a payload on the outer surface. Cowpea mosaic virus can be made to self-assemble into empty virus like particles in the absence of its RNA genome and can thus carry a payload both inside and outside of its protein shell.

In this review, we provide a series of examples to discuss how plant virus architecture contributes to their applications in cancer diagnostics and therapy [10].

## 2. Plant Virus Architecture

Virus size data that is found in different databases has been obtained by transmission electron microscopy (TEM), and virus sizes and architecture differs significantly [11]. A method of nanoparticle tracking analysis (NTA) can be used to determine the size and concentration of viruses and VLPs. At present, this technique has been used to study icosahedral VLPs of Cowpea mosaic virus (CPMV) (Figure 2) [12], spherical VLPs of structurally modified Tobacco mosaic virus (TMV) [13], and VLPs of filamentous Potato virus X (PVX) [14]. Nanoparticle tracking analysis has also been reported to compare the intensity of coated and native helical TMV [15,16].

Viruses are composed of outer protein shells that encapsulate the genomic material. The multiple copies of coat proteins that form the virus outer shell of viruses are collectively known as the capsid [17]. Primarily, the capsid occurs in different shapes and sizes and is meant to protect the genomic material to keep viruses safe under extreme environments [18]. Animal viruses are used as delivery vehicles for gene therapy, but the use of these viruses as nanocarriers can potentially create safety issues. Alternatively, plant viruses have not displayed toxicity and thus have received more attention for applications in nanomedicine [19]. The most well studied plant viruses are Tobacco mosaic virus (TMV), Cowpea mosaic virus (CPMV), and Potato virus X (PVX) (Figure 3) [20]. The immense diversity with respect to the shape and size of plant viruses enables them to be tailored for specific applications. For instance, rod shaped viruses are more suitable for penetrating into tumors [10]. The structural integrity of viruses remains intact even when surface properties have been altered through chemical and genetic modification; this allows control over targeting ligands, drugs and contrast agents for imaging [21]. The most interesting properties which give these plant based nanoparticles edge over synthetic nanomaterials include bulk production via host infection and available inexpensive large scale purification methods [4].

Researchers have the choice of selecting the most appropriate system for any given application owing to the diversity of plant virus shape and size. A wide range of nanomedical applications have been demonstrated using the capsids of icosahedral viruses [22]. Rod shaped viruses with a high aspect ratio have superior penetration into tumor tissue [10,23]. Various chemical and genetic approaches are reported to control the virus surface properties without affecting structural integrity, and that allow control on the attachment sites of drug molecules or contrast agents on the virus surface [21]. Plant-virus capsid pores are also reported to be employed to encapsulate small therapeutic molecules [24].

## 3. Categories of Plant Virus Nanoparticles

Plant virus-based NPs have received tremendous attention in recent years as providing the opportunity to target cancer cells. Tobacco mosaic virus (TMV) coat protein (CP) is one of the reported nanocarriers for a hydrophobic and insoluble peptide that targets the cancer cell receptor Neuropilin-1 (NRP1) [25]. TMV has been identified as a plant pathogen of the genus Tobamovirus and maintained status of a model as a plant virus for more than a century [26]. TMV has a genome size of 6400 bases, consisting of positive-sense single-stranded RNA. It is a rod-shaped virus with a mass of 39.6 MDa [27] and length and outer diameter of 300 and 18 nm respectively. Capsids of TMV are composed of identical 2130 protein subunits around the RNA right-handed helix to produce a 4 nm wide central hollow cavity [28]. The in vitro biochemical or genetic modifications in the TMV helical rod assembly system is possible due to its physical and chemical stability. There are several reports of modified TMV capsids to synthesize nanoparticles for use as nanocarriers in cancer treatment [29,30]. Antibodies, drugs and imaging agents can be conjugated to the external surfaces of TMV [31,32,33]. Virus-like nanoparticles (VLPs) derived from CP of TMV have been utilized by researchers as a suitable drug delivery nanocarrier [34] to load therapeutics, such as doxorubicin [35], phenanthriplatin [36], and mitoxantrone [37].

TMV can be used as a carrier of peptides with therapeutic or targeting activity against cancer. Trastuzumab is a cancer cell inhibiting monoclonal antibody that uses the binding sites of human epidermal growth factor receptor 2 (HER2). Trastuzumab-binding peptides (TBP) are immunogenic in nature and capable of initiating production of HER2-inhibiting antibodies to seize the growth of HER2-carrying cancer cells. TMV particles displaying TBP have been created to activate this immunogenicity [38].

A delivery system, PhenPt-TMV, has been reported whereby the anticancer drug phenanthriplatin was loaded into a hollow TMV carrier. It is an example of a stimuli responsive system, as the release of the drug is reported to be induced in the presence of an acidic environment [36].

The plant pathogen Cowpea mosaic virus (CPMV) belongs to the Comovirus genus. CPMV is an icosahedral shaped virus with a diameter of approximately 27 nm. It is composed of RNA-1 and RNA-2 of 6 and 3.5 kb, respectively, packed in 60 copies each of large and small coat protein [39]. CPMV is one of the most developed VNPs for biomedical and nanotechnology applications due to its ability to target specific tissues and act as an efficient drug delivery system. It is also reported to be well adapted for the attachment of a variety of molecules to the coat protein. Five reactive lysine residues of CPMV coat protein provide sites to chemically conjugate to various compounds; for instance: quantum dots [40], fluorescent dyes [41,42,43], polyethylene glycol (PEG) polymers [44] and various targeting moieties [45].

The icosahedral shape of CPMV capsid can be loaded with precise drug cargos to target tumor and cancer cells. Other qualities of CPMV include intravital imaging (imaging living cells while they are in a multicellular organism) and improved permeability with a retention effect that improves tumor penetration [46]. For in vivo imaging of tumors, CPMV-based VNPs have been successfully engineered to target specific tissues [45]. These tumor targeting VNPs also provide biocompatible platforms for cancer therapy and intravital imaging [46].

CPMV has been reported to interact with mammalian cells when administered orally or intravenously, or when a fluorescently modified form of virus is used to study intravital vascular imaging. The binding of CPMV to mammalian cells occurs through a cell-surface binding protein (CPMV-BP). Biochemical analysis of this CPMV-BP revealed the surface protein to be vimentin, a cytoskeletal protein known to modulate the interior of animal cells [47].

The CPMV-DOX conjugate was developed using eighty molecules of the chemotherapeutic drug doxorubicin (DOX), covalently bound to carboxylates at the external surface of the CPMV nanoparticle. This drug delivery vehicle was found to be more cytotoxic than free DOX when used in low concentration, however, CPMV-DOX cytotoxicity is time-delayed at higher concentrations [48]. Cancer cells manage to resist immunotherapies owing to the immunosuppressive nature of tumors. CPMV nanoparticles have been reported as an in situ vaccine to stimulate an anti-tumor response and overcome local immunosuppression [49]. CPMV has also shown to be effective for ovarian cancer. Such a strategy for immunotherapy resulting in antitumor efficacy is promising, and involves the formation of aggregates of CPMV and polyamidoamine generation 4 dendrimers (CPMV-G4). Administration of CPMV-G4 effectively reduced ovarian cancer [50]. CPMV nanoparticles thus provide a therapeutic application for tumor targeting, intravital imaging and cancer therapy [42].

As immunotherapies are resisted by tumor microenvironment, intratumoral administration of CPMV based nanoparticles as an immune stimulant has been reported to overcome the endogenous immunosuppression [49].

Potato virus X (PVX) is a member of the family Alphaflexiviridae, genus Potexvirus, and an important plant pathogen of the family Solanaceae that specifically infects potato, tomato and tobacco [51,52]. Among all of the proteinaceous plant virus nanoparticles reported for bioimaging, compared to TMV and CPMV, filamentous and biocompatible PVX has received slightly less attention. It has a 6.4 kb positive-stranded RNA genome [53]. Multiple copies of CP assemble around the genomic RNA to form the capsid. 

PVX can carry large payloads due to its flexible and filamentous structure, making it possible to use for pharmaceutical and imaging applications. Accordingly, PVX attain advanced tumor targeting and retention qualities, better than spherical nanoparticles. Qualities such as advanced pharmacokinetic profiles, low in vivo toxicity and enhanced functional properties make this protein-based nanoparticle suitable for biomedical applications [54].

The PVX particle is 515 × 14.5 nm in dimension and comprised of 1270 subunits of CP, 8.90 ± 0.01 per turn subunits and 3.45 nm of helix [55]. The C-terminus of each CP subunit is located internally and the N-terminus projected externally to the assembled particle, which provides a suitable site for modification [56]. The CP N-terminus is thought to keep the virus helical structure intact and therefore influence the stability of PVX particles.

When filamentous PVX VLPs are heated to 70 °C they swell, and at 90 °C, spherical VLPs are formed [57], akin to TMV structures at higher temperatures [58]. The maximum diameter of spherical PVX particles is 77 nm, these RNA free particles revealed different predicted secondary structures of denatured CP compared to the CP of the filamentous PVX particles [59]. 

Unlike other reported viruses, the assembly of PVX CP subunits into filamentous VLP, in vivo or in vitro, is not possible in the absence of genomic RNA. This reflects the unique connection between virus RNA and CP [60]. PVX nanoparticles produced so far have been used for external peptide presentation. With this limitation that the PVX particles can only be utilized in combination with genomic RNA, there are fewer reports regarding the internal channel of PVX CP carrying a drug payload or imaging molecule. Only the hydrophobic drug doxorubicin has been used to adhere to the grooves at the surface of PVX-based VNPs [61].

An efficient and new drug delivery system has been reported for Non-Hodgkin’s B cell lymphomas (NHL) based on PVX binding affinity towards malignant B cells. In a mouse model when PVX was loaded with the chemotherapy monomethyl auristatin (MMAE) and effectively administered to tissues harboring malignant B cells lead to inhibition of NHL growth. Thus, PVX based nanoparticle is an efficient drug delivery platform for malignancies of B cell [62].

Other plant VNPs are getting more attention due to their structural and functional characteristics when compared to synthetic nanoparticles. For example, the capsid proteins of plant virus Brome mosaic virus (BMV) have been reported for the synthesis of superparamagnetic nanoparticles (SPIONs) [45]. BMV is an icosahedral virus containing 4 ssRNA strands, RNA1 and RNA2 are packaged individually, while RNA3 and RNA4 are co-packaged. The capsid is composed of 180 identical proteins [63].

Viral capsids of icosahedral shaped Cowpea chlorotic mottle virus (CCMV) behave as an excellent natural protein cage for encapsulation and packaging of genomic nucleic acid. These molecular containers can be used as vessels for the entrapment and packaging of synthetic cargo [64]. Physalis mottle virus (PhMV) based VLPs have been used for the loading of drugs, such as, crystal violet, mitoxantrone, and doxorubicin; and dye molecules of photosensitizer and Zn-EpPor stably with the PhMV via noncovalent interactions. The cytotoxicity of these particles against various type of cancer cell lines has been confirmed [65].

The ricin epitope has been expressed on the surface of the icosahedral plant virus Tomato bushy stunt virus (TBSV) VLPs. Injection of these chimeric VLPs generated antisera into mice which makes TBSV a perfect choice for vaccine design [66].

The Virus-like nanoparticles (VLPs) derived from CP of plant viruses used to treat cancer are listed in Table 1.

## 4. Biomedical Applications of Plant Virus Nanoparticles

Magnetic resonance imaging (MRI) is a promising technology for the diagnosis of disease due to its high resolution and deep contrast, however, virus-based nanoparticles have been explored to overcome the drawback of low sensitivity [67]. TMV is reported as a carrier to deliver high payloads of MRI contrast imaging agents to sites of disease [68] and fluorescent dyes for biosensing and bioimaging [69]. This inert nature together with biological compatibility and multi-valency makes plant viruses suitable carriers of in vivo imaging agents. TMV rods have been conjugated to ‘BF3’, a multi-photon absorbing fluorophore which allowed mouse brain image over an extended duration without crossing the blood–brain barrier [70]. Ultra-high-field magnetic resonance imaging (UHFMRI) advances the diagnostic accuracy of MRI scans depending on better contrast agents for better resolution and signal-to-noise ratio. A bimodal contrast agent has been prepared to target integrin α2β1by loading the internal cavity of TMV nanoparticles with the complex of dysprosium (Dy3+) and the near-infrared fluorescence (NIRF) dye Cy7.5, and externally conjugated with an Asp–Gly–Glu–Ala (DGEA) peptide through a linker polyethylene glycol. This nanoparticle (Dy–Cy7.5–TMV–DGEA) was not only stable but also biocompatible with a low cytotoxicity. It achieved a high resolution when targeted to PC-3 prostate cancer cells [71].

Noninvasive imaging systems that are for early diagnosis of cancer, such as magnetic resonance imaging (MRI) and computed tomography (CT), have some limitations [72,73]. Nanotechnology offers an accurate state-of-the-art solution for imaging systems [72,73]. The capacity for multifunctionality and multivalency makes plant nanoparticle platforms an ideal choice for theranostic applications [46]. Nanoparticles are capable of precise molecular imaging to achieve accurate cancer diagnosis and therapy [74]. Delivery of imaging probes through nanostructures can improve the chances of early-stage cancer diagnosis through the use of multiple modalities to improve resolution, sensitivity, penetration, time, cost and on the top of all clinical relevance compared to the single imaging modalities [75]. Drug conjugated nanoparticles administered intravenously target tumors, via the process of enhanced permeability and retention (EPR) effect, depending on the type of tumor [76].

The use of nanoparticles (NPs) as drug delivery carriers for the treatment of infectious and chronic diseases including cancer are advantageous when compared with naked drugs [77]. The most promising nanoparticle systems have been adopted from naturally occurring plant viruses. Plant viruses are ideal for drug delivery as they are safe, non-infectious and nontoxic to humans [46]. Cancer cells exhibit specific antigens on the surface of tumor cells which can be identified and targeted by plant-virus based nanoparticles, thus providing a clinical application of diagnosis and therapeutics for cancer. The most promising nano-scale systems have been adopted from naturally occurring plant viruses such as Tobacco mosaic virus (TMV), Cowpea mosaic virus (CPMV), Potato virus X (PVX) and many more. Currently, these new strategies are only applied in small scale production. As these approaches undergo further development, we will witness a spectrum of possible applications in the fields of medicine and biomedical engineering.

VLPs mimic the native conformation of viruses, propagate their innate immunogenicity and promote safety due to the lack of their genetic material, and therefore form an attractive platform for vaccine development [78,79]. Furthermore, plant viruses are incapable of replication within mammalian cells, thus ensuring an additional level of safety during administration into tumors. VLPs serve as ideal vehicles for both prophylaxis (vaccine design) and therapy against cancer. VLPs are highly immunogenic and are readily phagocytosed by the antigen presenting cells (APCs), which in turn elicit antigen processing and display of pathogenic epitopes on their surfaces. As the VLPs are composed of multiple copies of their respective capsid proteins, they present repetitive multivalent scaffolds which aid in antigen presentation. Therefore, the VLPs prove to be perfect platforms for delivery and presentation of antigenic epitopes, resulting in induction of more a robust immune response compared to those of their soluble counterparts. As the tumor microenvironment poses the challenge of self-antigen tolerance, VLPs are preferrable platforms for delivery and display of self-antigens as well as otherwise weakly immunogenic antigens. The inherent multivalency and geometric arrangement of proteins on the VLP surface promote immune recognition via PAMPs (pathogen-associated molecular pattern receptors), most commonly Toll-like Receptors (TLRs) [80]. These properties, in addition to their diminutive size, enable the VLPs to deliver vaccines to the draining lymph nodes in addition to promoting APC interactions. Furthermore, many plant viral VLPs possess inherent adjuvant properties dispensing with the requirement of additional adjuvants to stimulate immune activity. Some of the highly immunogenic VLPs elicit innate immune activity, which in turn instigate adaptive immunity in tumor micro-environments [81]. Plant viral VLPs are nontoxic, inherently stable, and capable of being mass-produced as well as being modified with antigens and drugs, therefore provide an attractive option for eliciting anti-tumor immunity.

VLPs have been attached to TAAs (tumor associated antigens) using chemical conjugation, genetic fusion and enzyme-mediated ligation techniques. Among these various strategies, the most common method has been chemical conjugation. For example, the human epidermal growth factor receptor 2 (HER2) epitope, when conjugated to the icosahedral CPMV, was successfully delivered to the lymphatic system with enhanced uptake and activation of APCs that led to an augmented anti-HER2 immune response. The CPMV HER2 candidate vaccine slowed tumor progression and metastasis in mouse models, enhancing survival [82]. Importantly, CPMV-HER2 stimulated a predominantly Th1 immune response while Sesbania mosaic virus-HER2 and CCMV-HER2 induced mostly a Th2 response in mouse models, thus proving that the nature of the epitope carrier itself plays an essential role in regulating the Th1/Th2 bias. This could be due to differences in epitope display on the surface of the VNPs as well as the capsid. In another study, when PVX was conjugated to a recombinant idiotypic (Id) TAA displayed a 7-times greater anti-Id IgG response compared to that elicited by Id alone in a mouse B-cell lymphoma model (BCL1) [83]. It was determined that TLR7 was crucial for the recognition of the viral RNA, in addition, cytokines such as IL-12 and IFN-α were induced.

Cancer vaccines against carbohydrate antigens associated with tumors (TACAs) could be useful for diminishing tumor progression. Nevertheless, carbohydrates are weakly immunogenic and therefore, plant viruses used as carriers of these molecules could enhance the immune response to TACAs. CPMV-TACA conjugates targeting the Tn antigen (GalNAc-α-O-Ser/Thr) [84] were demonstrated to induce enhanced IgG titers, implicating heightened T-cell mediated immunity and antibody isotype switching in mouse models. IgG binding to the Tn antigens were observed in experiments wherein mice sera were added to breast cancer cell lines. In another subsequent study, TMV was used as an equivalent system to display Tn antigens [85] in which conjugation to Tyr 139 of TMV generated robust immune responses. Finally, nucleic acid vaccines have become recognized as next-generation vaccines against cancers, and VLPs are currently being investigated as potential carriers. Nanocarriers such as plant VLPs could be used to protect RNA-based vaccines, thereby extending the half-lives and biodistribution of these therapeutics. 

The tumor microenvironment poses a great challenge to immune clearance by virtue of being immunosuppressive and favoring immune escape of the tumors through the inhibition of anti-tumor T-cells [79,86]. CPMV VLP nanoparticles were shown to decrease tumor growth in murine models of lung melanomas, ovarian, colon and breast tumors [49,87,88]. Mechanistically, CPMV has been shown to reprogram the tumor microenvironment by recruitment of natural killer cells and neutrophils, while enabling the transition of M2 to M1 anti-tumor macrophages. This innate immune cell population subsequently combats the tumor leading to cell lysis while allowing the antigen presenting cells to process and display the tumor associated antigens resulting in strong T-cell activation and systemic immunity against the tumors. CPMV VNPs have also been formulated as slow-release aggregates along with polyamidoamine generation 4 dendrimers (CPMV-G4) [50], where they were shown to be effective in combating ovarian cancer in murine models, even when provided as a single dosage. Besides CPMV, TMV and PapMV have also been shown to have anti-tumor potential, although CPMV proved to be more effectual.

Plant virus VLPs have also been used as combination therapies to augment their immune efficacy. The PVX-DOX (doxorubicin) [61] combination was shown to be highly effective in stimulating cytokine/chemokine levels while prolonging the survival of mice in melanoma models when compared to that obtained through the administration of either PVX or DOX alone. Moreover, the strong chemotherapeutic cyclophosphamide, when used in combination with CPMV VNPs, profoundly elicited tumor cell death, releasing extracellular TAAs and stimulating immune cell invasion in addition to augmenting TAA recognition and antigen presentation [49] in mouse tumor models. CPMV VNPs have also been administered as combinations with CD47-blocking antibodies [89] which proved to have synergistic effects in combating tumor growth in murine ovarian tumor models, where it activated phagocytes. The anti-CD47 antibodies suppressed antiphagocytic signals, leading to stimulation of the adaptive immune response. Similar synergistic effects were observed when CPMV VNPs were used in combination with the anti-programmed cell death-1 checkpoint inhibitor. In addition to this, CPMV has been used successfully in promoting anti-tumor effects, when combined with radiation therapy. In this instance, CPMV was shown to enhance the recruitment of APCs, which in turn targeted the extracellular TAAs and phagocytosed them to induce a prolonged effectual immune response [65] in mice and canine models. 

Presently, clinical treatment for cancer is primarily addressed by chemotherapy [71]. Nevertheless, the high recurrence and resistance rates as well as the fast clearance of anti-cancer drugs and non-targeted drug delivery necessitate the administration of maximum tolerable doses of drugs for enabling cancer therapy. This may lead to increased toxicity and diminished clinical pertinence. Therefore, drug delivery technologies that enable intracellular and targeted conveyance while promoting active drug accumulation in tumors, in concert with the limiting of dose requirements, could alleviate these concerns and augment treatment outcomes. 

Plant virus VLPs have several attractive features that make them appropriate for targeted administration of therapeutic molecules. The anti-cancer drug doxorubicin (DOX), has been successfully delivered using VNPs and VLPs. Specifically, red clover necrotic mosaic virus (RCNMV) VLPs have been used to both encapsulate DOX [90] as well as bind DOX on its surface. This enabled the simultaneous quick release of the externally-bound DOX and the slow release of the encapsulated DOX from within the VLP. Similarly, TMV- and PVX-derived VLPs and VNPs have been successfully used to deliver DOX [91], which proved to be highly efficacious. In this context, molecules with high aspect ratio have proven to be of great use in effective drug delivery, of which helical plant viruses that form tubular and filamentous structures are good representative examples. VNPs have shown great promise as their cargo-RNA functions as a ruler establishing the length of the virus particle and simple adsorption of DOX on their surface was shown to be effective for reducing tumor growth [3,35].

Plant virus VNPs have been used for targeted administration of platinum-based drugs against cancer. This is important, as 50% of chemotherapy treatments involve the use of these platinum-derived drugs. TMV has been demonstrated to efficiently deliver Cisplatin [92] and Phenanthriplatin [93], both of which are platinum-based drugs. The drugs were loaded into the TMV VNP cavity using charge-driven interactions or by synthesizing stable covalent adducts. Such a TMV-based drug delivery system was proven to enable superior, targeted cytotoxicity as well as increased ease of uptake by cancer cells in in vitro systems using HepG2 and MCF-7 cancer cell lines [34]. 

Another anti-cancer drug, mitoxanthrone (MTO), is a topoisomerase II inhibitor and has been shown to be encapsulated by TMV [37] and CPMV [94] VNPs exhibited superior tumor-reduction in mouse cancer models, while precluding severe cardiac outcomes that sometimes accompany direct delivery of MTO. Yet another anti-neoplastic and antimitotic drug, valine-citrulline monomethyl auristatin E (vcMMAE), was bound to the exterior of TMV VNPs which targeted non-Hodgkin’s lymphoma. Internalization into endolysosomal compartments was reported [95], most likely accompanied by the protease-mediated release of the drug. This system was efficient in terms of cytotoxicity towards the in vitro Karpas 299 non-Hodgkin’s lymphoma cell line with an IC50 of 250 nM.

Viruses are inherently natural gene-delivery vehicles. Plant viral VLPs in particular, by virtue of being devoid of their own genetic materials, easily encapsidate nucleic acids. This makes them highly useful in delivering genes and therapeutic nucleic acids. It is reported that (Gene Therapy Clinical Trials Worldwide, 2019) 67% of clinical trials involving gene therapy use VLPs and viruses as vectors for transmission of genetic materials. Tian et al., (2018) demonstrated that transacting activation transduction (TAT) peptide conjugated to the external surface of TMV augmented internalization along with increased ability to escape endo/lysosomal compartments [96]. Most of these VLPS exhibited uptake by dendritic cells and macrophages and proved to be highly immunogenic. Thus, therapeutic nucleic acids can be easily delivered to immune cells during cancer treatments. The CpG oligodeoxynucleotide (ODN) was loaded onto CCMV VLPs which greatly increased uptake by TAM (tumor associated macrophages), not by cancer cells. Intra-tumoral delivery of CpG-loaded CCMV VLPs demonstrated significant inhibition of tumor growth in mouse models [49].

In comparison to nucleic acids, proteins and polypeptides are more difficult and complex to encapsulate, thus limiting their delivery into cancer cells. This could be resolved by fusing the polypeptide to the VLP capsid protein by chemical or genetic means. This could direct cargo localization into the interior of the VLPs during the synthesis and assembly of the capsid in vivo [97]. Viral capsid protein could spontaneously self-assemble around the polypeptide in the presence or absence of a targeting element in vitro [98]. Several drug-activating enzymes encapsulated within VLPs enable therapeutic functions [99]. Amongst these, the most important are the cytochrome P450 family of enzymes which catalyze into active forms of their precursors, namely the chemotherapeutic pro-drugs. This enables better tumor-targeting, enhancing their half-life within the tumors via better permeability while reducing side effects [100,101]. In this context, the CCMV has been shown to encapsulate cytochrome CYPBM3 with demonstrated activation effects on the pro-drugs resveratrol and tamoxifen [100]. Additionally, these proteins can be displayed on the VLP surfaces for therapeutic or targeting purposes.

Helical viruses such as PVX [102] and TMV are ideal vehicles for drug delivery by virtue of their shape and high aspect ratio that enables augmented cellular interactions with their conjugated cargo. PVX displaying TNF related apoptosis inducing ligand (TRAIL) was used to promote the recruitment and activation of death receptors in vitro in HCC-38 primary ductal carcinoma, BT-549 ductal carcinoma and the MDA-MB-231 breast cancer cell lines [61,102]. In vivo mouse models also demonstrated that the PVX-TRAIL formulation potently inhibited tumor growth.

## 5. Challenge and Future Directions

VNPs developed from plant viruses are non-infectious for warm-blooded mammals, comprised of only virus shells and lacking a viral genome. Genetically engineered viruses can be selected for required binding sites and modified properties [103]. In addition, stimulus-sensitive nanoparticles are developed knowing the drug delivery mechanisms and the expected therapeutic outcomes. The possibilities of application for plant viruses and virus-like particles as VNPs are unlimited and depend upon the anticipated advantages of these biomolecules. The understanding of nucleoprotein complexes consisting of vast quantities of identical coat protein subunits has facilitated the use of plant virus nanoparticles for several decades [104]. Plant virus-based nanotechnology makes use of Cowpea mosaic virus (CPMV), Cowpea chlorotic mottle virus (CCMV), Potato virus X (PVX), and Tobacco mosaic virus (TMV) [17]. The viral coat protein can be tailored according to need with functional ligands and structures for applications such as molecular imaging or therapy [105]. 

The future of plant virus nanoparticles in anti-cancer therapy will continue to evolve. The vast majority of scientific studies have taken place either in cell culture or in animal models. Thus, the next significant challenge will be the design and implementation of human clinical trials. The authors are unaware of any such trials that have been undertaken to date; however, much talk of this next step is ongoing within the scientific community, with optimistic expectations with respect to safety and efficacy. Besides this initial hurdle, delivery mechanisms for plant virus particles into patients need to be explored in more detail. Finally, a regulatory framework must be set in place to ensure that plant virus nanoparticles can safely reach the market.

## Figures and Tables

**Figure 1 vaccines-09-00830-f001:**
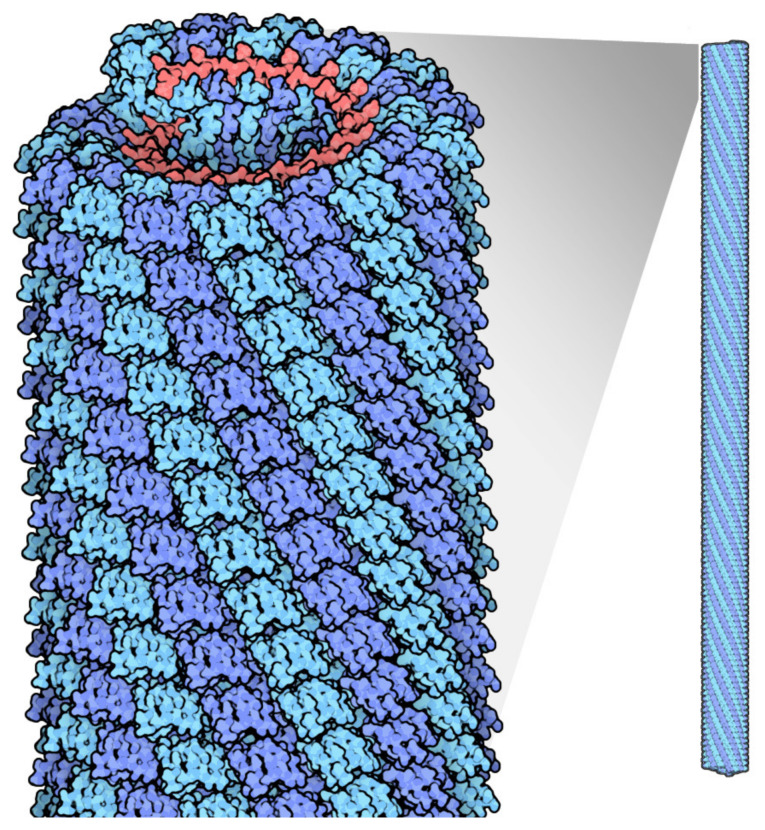
Tobacco mosaic virus structure, RNA is in red, protein subunits in blue. Source: https://pdb101.rcsb.org/motm/109 (accessed on 11 May 2021).

**Figure 2 vaccines-09-00830-f002:**
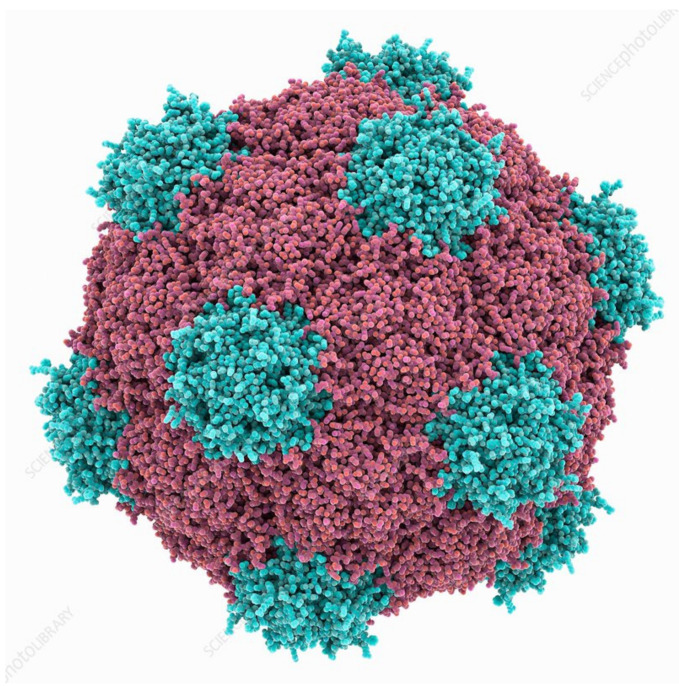
Cowpea mosaic virus structure, Protein subunits in red and blue. Source: fineartamerica.com (accessed on 11 May 2021).

**Figure 3 vaccines-09-00830-f003:**
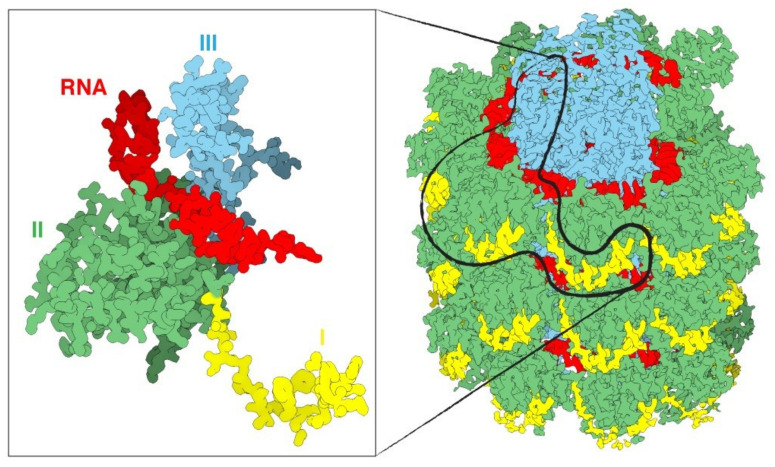
An overview of a portion of the PVX virus (right). The three domains of the protein are shin in yellow, green and cyan, the RNA in red. The magnification on the left displays only one single CP with a fragment of RNA. Source: https://naturemicrobiologycommunity.nature.com/posts/61805-a-detailed-view-of-a-filamentous-virus-potato-virus-x (accessed on 11 May 2021).

**Table 1 vaccines-09-00830-t001:** Virus-like nanoparticles (VLPs) derived from CP of plant viruses used to treat cancer.

S. No.	Plant Virus	Shape of Virus	Genome	Use of VLPs	References
1	Tobacco mosaic virus (TMV)	Rod-shaped	Single-stranded RNA (6.4 kb)	Target Neuropilin-1 (NRP1)	[25]
				Load therapeutics, such as doxorubicin, phenanthriplati, and mitoxantrone	[35,36,37]
				Carrier of Trastuzumab-binding peptides (TBP)	[38]
2	Cowpea mosaic virus (CPMV)	Icosahedral-shaped	RNA-1 (6 kb) and RNA-2 (3.5 kb)	Target Ovarian Cancer Cells	[50]
				Bound to carboxylates of doxorubicin at the external surface of nanoparticle.	[48]
				In vivo imaging of tumors	[45]
3	Potato virus X (PVX)	elongated filaments	Single-stranded RNA (6.4 kb)	doxorubicin adheres to the surface of PVX-based VNPs	[61]

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
