# Peer review of "Frontiers in Bioengineering and Biotechnology: Plant Nanoparticles for Anti-Cancer Therapy"

_vaccines, 2021, doi:10.3390/vaccines9080830_

Round 1

Reviewer 1 Report

  1. The manuscript could benefit from minor editing for grammar, missing words, and subject-verb agreement, etc. It is recommended that authors delete irrelevant "general" phrases and sentences, repeated and unneeded words. They should use short sentences. Also, some Introductory sentences are irrelevant or are not needed.
  2. The title could be improved by replacing the period with a colon.
  3. All abbreviations should be revised and defined at their first use. In the abstract, the authors used VNPs to abbreviate plant virus nanoparticles then used VLPs. What do VLPs mean? And why is the abbreviation of plant virus nanoparticles “VNPs” and not “PVNs”?
  4. Abstract: It is well-written, but authors should start by defining what plant virus nanoparticles are and their applications before introducing the idea of using them for cancer therapeutics.
  5. Introduction: The authors are advised to start the introduction with defining VNPs and their applications then tackling the topic of challenges facing cancer therapeutics. Then, they can emphasize on the need for new technologies such as VNPs in cancer. I am also wondering what is the difference between plant and animal virus nanoparticles?
  6. Introduction: I suggest that authors add a paragraph to the introduction talking about cancer therapeutics and role of nanoparticles in improving management of patients.
  7. Authors can add tables to facilitate and summarize main ideas included. For instance, a table can be included that list all “categories of plant virus nanoparticles.
  8. It would be nice also to add a figure demonstrating the use of VLPs and VNPs in cancer.
  9. More subheadings should be added rather than just narrating ideas under one heading of “Categories of plant virus nanoparticles.” I would suggest adding the different applications in cancer under various subheadings.
  10. The whole manuscript should be revised, references should be updated.

Author Response

  1. The manuscript could benefit from minor editing for grammar, missing words, and subject-verb agreement, etc. It is recommended that authors delete irrelevant "general" phrases and sentences, repeated and unneeded words. They should use short sentences. Also, some Introductory sentences are irrelevant or are not needed. We changed the grammar and wording of the manuscript.
  2. The title could be improved by replacing the period with a colon. Thanks for the suggestion.
  3. All abbreviations should be revised and defined at their first use. In the abstract, the authors used VNPs to abbreviate plant virus nanoparticles then used VLPs. What do VLPs mean? And why is the abbreviation of plant virus nanoparticles “VNPs” and not “PVNs”? We have made these changes.
  4. Abstract: It is well-written, but authors should start by defining what plant virus nanoparticles are and their applications before introducing the idea of using them for cancer therapeutics. Thanks, we have revised as suggested.
  5. Introduction: The authors are advised to start the introduction with defining VNPs and their applications then tackling the topic of challenges facing cancer therapeutics. Then, they can emphasize on the need for new technologies such as VNPs in cancer. I am also wondering what is the difference between plant and animal virus nanoparticles? Thanks, we have now revised.
  6. Introduction: I suggest that authors add a paragraph to the introduction talking about cancer therapeutics and role of nanoparticles in improving management of patients. Thanks, we have revised.
  7. Authors can add tables to facilitate and summarize main ideas included. For instance, a table can be included that list all “categories of plant virus nanoparticles.” Thank you for your comment, a table has been added.
  8. It would be nice also to add a figure demonstrating the use of VLPs and VNPs in cancer. A figure has been added.
  9. More subheadings should be added rather than just narrating ideas under one heading of “Categories of plant virus nanoparticles.” I would suggest adding the different applications in cancer under various subheadings. We have revised as suggested.
  10. The whole manuscript should be revised, references should be updated. We have completed this.

Reviewer 2 Report

This is an interesting and timely review of the use of plant virus particles as a means of combating cancer. The authors cover the main viruses used for this purpose in a comprehensive manner.

My main comment about the manuscript is the lack of display items, particularly figures. The authors described the advantages of plant virus particles in terms of their size and shape but these are not illustrated in any way. Thus readers would really have to be already familiar the viruses in question to get the most out of the review. I would urge the authors to include some figures.

Another, more minor, issue concerns the references cited when describing the fundamentals of the various viruses. These tend to be from very recent publications describing the use of the particles and do not really enable the reader, should they wish, to find out more about the underlying virology. I would prefer to see references to review articles about the viruses or even web addresses where additional information can be found.

Author Response

This is an interesting and timely review of the use of plant virus particles as a means of combating cancer. The authors cover the main viruses used for this purpose in a comprehensive manner.

My main comment about the manuscript is the lack of display items, particularly figures. The authors described the advantages of plant virus particles in terms of their size and shape but these are not illustrated in any way. Thus readers would really have to be already familiar the viruses in question to get the most out of the review. I would urge the authors to include some figures. We have added a figure.

Another, more minor, issue concerns the references cited when describing the fundamentals of the various viruses. These tend to be from very recent publications describing the use of the particles and do not really enable the reader, should they wish, to find out more about the underlying virology. I would prefer to see references to review articles about the viruses or even web addresses where additional information can be found. We have revised the references.